# Predicting Time-varying Flux and Balance in Metabolic Systems using Structured Neural ODE Processes

## Abstract

We develop a novel data-driven framework as an alternative to dynamic flux balance analysis, bypassing the demand for deep domain knowledge and manual efforts to formulate the optimization problem. The proposed framework is end-to-end, which trains a structured neural ODE process (SNODEP) model to estimate flux and balance samples using gene-expression time-series data. SNODEP is designed to circumvent the limitations of the standard neural ODE process model, including restricting the latent and decoder sampling distributions to be normal and lacking structure between context points for calculating the latent, thus more suitable for modeling the underlying dynamics of a metabolic system. Through comprehensive experiments (156 in total), we demonstrate that SNODEP not only predicts the unseen time points of real-world gene-expression data and the flux and balance estimates well but can even generalize to more challenging unseen knockout configurations and irregular data sampling scenarios, all essential for metabolic pathway analysis. We hope our work can serve as a catalyst for building more scalable and powerful models for genome-scale metabolic analysis.

## 1 Introduction

A distinctive characteristic of deep neural networks is their capability to implicitly learn complicated features and dynamics from data, significantly saving human effort in composing those handcrafted features and devising complex models. Therefore, there has been a growing interest in using them in a variety of scientific contexts, such as quantum chemistry (von Glehn et al., 2022), tokamak controller design (Degrave et al., 2022), climate sciences (Lam et al., 2022; Nguyen et al., 2023), molecule generation (Hoogeboom et al., 2022) and drug discovery (Askr et al., 2023), to name a few. For drug discovery problems in particular, it is essential to answer the questions of where and how the drug should be targeted. The machine learning community has attracted increased attention in molecular design to address the latter question (Luo et al., 2022; Corso et al., 2022). On the other hand, metabolic pathway analysis techniques, such as flux balance analysis (FBA) (Orth et al., 2010) and dynamic FBA (Mahadevan et al., 2002), have been shown highly effective in finding drug targets (Sen & Orešič, 2023). These methods are widely used to study the effect of drugs or environmental stress simulated by gene knockouts on unwanted cells, such as cancer cells, by curbing their metabolism (Raškevičius et al., 2018). Nevertheless, several key parameters, including the optimization objective and constraints for the reaction flux in their linear programming (LP) formulation, must be determined using domain expertise for each case, largely limiting their generality and scalability. In this work, we aim to develop scalable data-driven methods that can directly predict the behavior of metabolic systems with time-varying flux, thus avoiding the manual effort required to build FBA models.

More specifically, we achieve this by leveraging single-cell RNA sequencing (scRNA-seq) time-series data (Chen et al., 2019) and using single-cell flux estimation analysis (scFEA) technique from Alghamdi et al. (2021) to estimate flux and balance of the metabolic system, because scRNA-seq can churn out data in bulk, and getting time-series single-cell gene-expression data is much less labor intensive than getting actual flux-balance time-series data. The challenge, however, lies in that gene expression trajectories for individual cells cannot be tracked over time since cells die once their gene expression is read. Instead, we only have gene expression samples from different cells at each

timestep, which can be viewed as samples from a time-varying distribution resembling a random process. In fact, it's well known that gene transcription is stochastic, especially when considered at the single cell level (Thattai & Van Oudenaarden, 2001). Thus, the amounts of molecules produced, or the chemical concentration, from a collection of cells can be considered to be sampled from some distribution, with the amounts of mRNA molecules showing a Poisson-like behavior in a steady state as shown in Thattai & Van Oudenaarden (2001).

Since the time-varying metabolic concentrations are known to follow a non-linear ordinary differential equation (ODE), we propose a novel *Structured Neural ODE Process* (SNODEP) architecture that is built on top of the standard neural ODE processes (Norcliffe et al., 2021) to predict the underlying dynamics of the metabolic system. We note that standard neural ODE processes have several design choices that might not help to model the ODE dynamics in our case, like lack of structure in the encoder to get the latent distribution from the context points and the use of Gaussian parametric family for latent posterior and decoder distributions. Consequently, we design the architecture of SNODEP to bypass these shortcomings, showing improved performance in tasks such as predicting gene-expression distributions on unseen timesteps, predicting metabolic-flux and metabolic-balance distribution on unseen timesteps, and predicting the corresponding distributions for gene-knockout cases, considering both regularly and irregularly sampled data, all for several metabolic pathways.

**Contributions.** We formulate the prediction problem of metabolic flux and balance as a stochastic neural processing task, where the goal is to learn the underlying dynamics by predicting their time-varying distributions under different configurations (Section 2). We propose an end-to-end training framework, which first defines the intermediary steps required to estimate metabolic flux and balance from scRNA-seq data and then learns a novel SNODEP model that can predict the unseen time points of flux and balance and their dynamics under gene-knockout configurations (Section 3.2). The proposed SNODEP architecture is designed by addressing a few limitations of the standard architecture of neural ODE processes (Section 3.1); thus, it is more suitable to model the time-varying distributions from metabolic systems. Comprehensive experiments on real-world datasets and various metabolic pathways demonstrate that SNODEP is highly effective in modeling the dynamics of gene expressions and predicting metabolic flux and balance, consistently outperforming alternative models such as standard neural ODE processes (Sections 4.2-4.4). We also showcase the superiority of SNODEP under gene-knockout variations and scenarios with irregularly sampled data (Section 4.5), suggesting its versatility and strong potential in solving challenges in biomedical domains.

## 1.1 RELATED WORK

**Metabolic Pathway Analysis.** Genome-scale metabolic models (GSMMs) have proven to be powerful tools in the design of therapeutic treatments. For instance, Raškevičius et al. (2018) employed GSMMs to identify therapeutic windows for cancer treatment, while Larsson et al. (2020) used them to simulate gene knockouts in a Glioblastoma cancer cell model, identifying potential therapeutic targets and predicting side effects in healthy brain tissue. Despite their importance, GSMMs are time-consuming and require significant domain expertise to build. Recent studies have explored integrating machine learning techniques with GSMMs, as reviewed in Sahu et al. (2021). From a dynamical standpoint, Costello & Martin (2018) framed pathway dynamics prediction as a machine learning problem, using XGBoost models to predict such dynamics, but their framework is not end-to-end. More recently, Aghaee et al. (2024) introduced a graph neural network model to simulate the dynamic behavior of metabolites in oxidative stress pathways in bacterial cell cultures for synthetic data. In addition, RNA velocity (La Manno et al., 2018) estimates the time derivative of gene expressions but needs spliced and unspliced mRNA counts, usually not reported in the experiments. Similarly, Klumpe et al. (2023) investigated single-cell time series prediction, albeit also using synthetic data with no specific focus on metabolic pathways. To the best of our knowledge, our work is the first to comprehensively study the dynamically varying flux and balance of metabolic pathways derived from real-world single-cell gene expression time-series data.

**Neural ODE.** The neural ODE family of models has shown strong capabilities in modeling dynamic systems, particularly when the underlying dynamics are known to follow an ODE (Rubanova et al., 2019). While latent neural ODEs have been applied to interpolation and extrapolation tasks, they are not suitable for modeling random processes. In contrast, neural processes (NP) (Garnelo et al.,

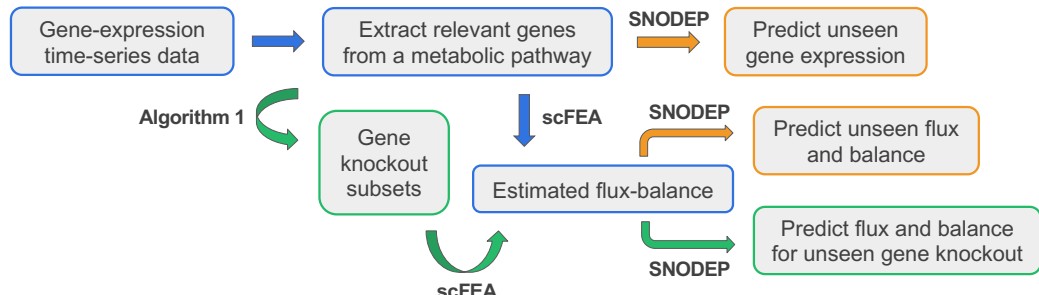

Figure 1: Overall pipeline of our framework for predicting time-varying distributions, such as gene expressions, flux, and balance, with (green) and without (orange) gene knockouts.

2018) can be used for modeling time-varying distributions, but they have no consideration for the underlying dynamics. These observations motivate us to explore models like neural ODE processes (NODEP) (Norcliffe et al., 2021), where the dynamics are defined over the parametric space of these distributions. Other models, such as those proposed in Kidger et al. (2021), assume a noisy evolution of dynamics, which does not align with our prediction problems of time-varying distributions in metabolic systems. Our work adapts standard neural ODE processes (Norcliffe et al., 2021) to better suit our specific settings, showing improvements across various tasks and metabolic pathways.

## 2 PROBLEM FORMULATION

Classical methods like DFBA estimate time-varying metabolic flux and balance by solving an optimization problem to maximize the biomass at each timestep (see Appendix B.1 for more details). Our work proposes to directly train a model on scFEA-estimated flux-balance values until a certain timestep and then predict the distributions of gene expression, flux, and balance in future timesteps, expecting that the trained model will learn the underlying dynamics. Figure 1 illustrates the overview of our pipeline. Due to page limits, we defer more details on scFEA proposed by Alghamdi et al. (2021) to Appendix B.2. Below, we provide detailed descriptions of our problem setup. The key notations and their descriptions are provided in Appendix A.

**Predicting Gene Expression, Flux and Balance.** Suppose we have a gene count matrix of dimension $K \times N$, where $N$ is the total number of cells and $K$ is the total number of genes, with gene counts measured at each *regular* timestep $t$ and total $V$ timesteps. Let $\mathbb{B}_t$ be the index set representing the cells whose gene counts $\in \mathbb{R}^K$ are observed at time $t$. Then, we have $\sum_t |\mathbb{B}_t| = N$, indicating that all $N$ cells get their expressions counted over various timesteps.

For a metabolic pathway, we only extract the relevant $d$ genes from the total set of $K$ genes. Let $g_{i,t} \in \mathbb{R}^d$ be the gene-expression array for cell $i \in \mathbb{B}_t$ at time $t$, and $\mathbf{G}_t \in \mathbb{R}^{d \times |\mathbb{B}_t|}$ be the corresponding matrix. For a certain metabolic pathway with $u$ modules and $v$ metabolites and each batch $\mathbb{B}_t$ of cells at time $t$, we estimate the flux $m^f$ and balance $m^b$ using the scFEA framework detailed in Appendix B.2. Specifically, we define:

- $S_t^f : \{g_{i,t}\}_{i \in \mathbb{B}_t} \rightarrow \{m_{i,t}^f\}_{i \in \mathbb{B}_t}$ as the mapping that estimates the flux $m_{i,t}^f \in \mathbb{R}^u$ for each cell $i$ based on its gene expression. Let $\mathbf{M}_t^f \in \mathbb{R}^{u \times |\mathbb{B}_t|}$ be the matrix of the flux samples.
- $S_t^b : \{g_{i,t}\}_{i \in \mathbb{B}_t} \rightarrow \{m_{i,t}^b\}_{i \in \mathbb{B}_t}$ as the analogous mapping for estimating the metabolic balance $m_{i,t}^b \in \mathbb{R}^v$. Let $\mathbf{M}_t^b \in \mathbb{R}^{v \times |\mathbb{B}_t|}$ be the matrix of the balance samples.

We note that scFEA was originally developed for static-FBA, but since the static-DFBA formulation (Equation 4) can be interpreted as solving the static-FBA for different timesteps, we use scFEA to estimate flux-balance values for different timesteps.

**Gene-knockout.** Gene knockout is a way to understand how a gene influences the metabolic network, for example, in understanding how essential genes in pathogens affect metabolic pathways to design drugs to inhibit those pathways (Larsson et al., 2020); it's also widely used in synthetic

biology Dalvie et al. (2021). In the gene-knockout simulations in FBA models, the constraints of the reaction fluxes affected by essential genes are usually modified (Maranas & Zomorrodi, 2016). In contrast, we train a model on certain gene-knockout configurations and then predict the distribution on unseen configurations and timesteps. To simulate gene-knockout conditions, we randomly sample $S$ subsets of $k$ most expressed genes, set the gene-expression of genes from those subsets to zero (See Algorithm 1 for more details), and estimate the flux-balance values again based on the scFEA techniques. For $s \in \{1, 2, \ldots, S\}$, we analogously define $\{\tilde{m}^f_{i,t}\}_{s,i\in\mathbb{B}_t}$ and $\{\tilde{m}^b_{i,t}\}_{s,i\in\mathbb{B}_t}$ as gene-knockout flux and balance estimates, respectively, where we use $\tilde{\mathbf{M}}^f_{s,t} \in \mathbb{R}^{u\times|\mathbb{B}_{s,t}|}$ and $\tilde{\mathbf{M}}^b_{s,t} \in \mathbb{R}^{v\times|\mathbb{B}_{s,t}|}$ to denote the corresponding matrix of samples.

Essentially, our framework assumes that metabolic flux and balance from scRNA-seq data can be estimated using scFEA techniques, that knocking out a subset of genes does not change the expression levels of the rest of the genes, and that gene essentiality and gene expression levels are correlated.

**Learning Objective.** For each timestep $t \in \{t_1, t_2, \ldots, t_V\}$, we collect samples of gene expression $\{g_{i,t}\}_{i\in\mathbb{B}_t}$, flux $\{m^f_{i,t}\}_{i\in\mathbb{B}_t}$ and balance $\{m^b_{i,t}\}_{i\in\mathbb{B}_t}$ and their gene-knockout samples $\{\{\tilde{m}^f_{i,t}\}_{j,i\in\mathbb{B}_t}, \{\tilde{m}^b_{i,t}\}_{j,i\in\mathbb{B}_t}\}$ with cells $\mathbb{B}_t$ using previous steps. We assume these samples are drawn from some underlying distributions corresponding to gene expression $G(\theta_g(t))$, flux $M^f(\theta_f(t))$, balance $M^b(\theta_b(t))$ and their gene knockout versions $\{\tilde{M}^f(\theta_f(t)), \tilde{M}^b(\theta_b(t))\}$, respectively. The goal is to learn a model $F : t \to Y(\theta_t)$ that can predict the underlying dynamics of time-varying distributions, which depend on some latent distribution $L$. In our setup, $F$ is considered as an encoder-decoder neural network, with a different network for each distribution in $\{G, M^f, M^b, \tilde{M}^f, \tilde{M}^b\}$.

Let $C < T < V$ be the length of context, target, and total available time points, respectively. Given a distribution $Y$, let $y_i \sim Y(\theta_t)$ for any $i \in \{1, \ldots, V\}$. When we say $y_i \sim Y(\theta_t)$, it means a random sample from the set $\{y_{i,t}\}_{i\in\mathbb{B}_t}$. During training, our model's encoder takes as input the context data, which includes samples from context points $\mathcal{C} = \{(t_1, y_1), \ldots, (t_C, y_C)\}$. The decoder then predicts samples from the target points $\mathcal{T} = \{(t_1, y_1), \ldots, (t_T, y_T)\}$. During inference, the model is used to predict every timestep available, including hitherto unseen timesteps $\mathcal{V} = \{(t_1, y_1), \ldots, (t_V, y_V)\}$. In the following discussions, we denote $\mathbb{I}_\mathcal{C} = \{1, \ldots, C\}$ and $\mathbb{I}_\mathcal{T} = \{1, \ldots, T\}$ for simplicity.

## 3 METHODOLOGY

### 3.1 ISSUES WITH STANDARD NEURAL ODE PROCESS

The standard neural ODE process (NODEP) model (Norcliffe et al., 2021) employs an encoder-decoder model architecture, where the context points $\{(t_i, y_i)\}_{i\in\mathbb{I}_\mathcal{C}}$ are used to calculate the latent distributions $L_0(\theta_{l_0})$ and $D(\theta_d)$, and the latent $l_0 \sim L_0$ evolves over target timesteps $\{t_i\}_{i\in\mathbb{I}_\mathcal{T}}$, according to an ODE that is modeled by a neural network $\mathbf{f}_w$ as follows:

$$l(t_i) = l_0 + \int_{t_0}^{t_i} \mathbf{f}_w(l(t), d, t)dt. \tag{1}$$

The time-evolving latent distributions are then fed into a decoder to obtain the target distributions: $\{N_i(y_i|\mu_{w_1}(l(t_i)), \sigma_{w_2}(l(t_i)))\}_{i\in\mathbb{I}_\mathcal{T}}$. Although NODEP has been shown effective in modeling ODE dynamics for scientific discovery, there are a few limitations with NODEP if applied to our settings:

1. The latent and decoding distributions are treated as Normal. This is not the best choice of distributions to model gene-expression data, which is usually discrete and Poisson-like (Thattai & Van Oudenaarden, 2001) and confirmed by Figures 9a and 9b in Appendix E.

2. The encoded representation $r_i = f_e(\{t^\mathcal{C}_i, y^\mathcal{C}_i\})$ is calculated using context points without any particular structure in NODEP. These $r_i$'s are then order-invariantly aggregated to give $r$, and finally $D \sim q_D(d|\mathcal{C}) = \mathcal{N}(d|\mu_D(r), \text{diag}(\sigma_D(r)))$, similarly for $L_0$. The order between the context points and their sequential dependence on each other is not efficiently utilized. Enforcing this sequential dependence can be highly useful for guiding the ODE decoder because otherwise, it might lead to unintended attention to certain context points.

This sequential dependence of context points is even more important for irregularly sampled data, where an order-invariant encoder might lead to different representations for different timesteps sam-

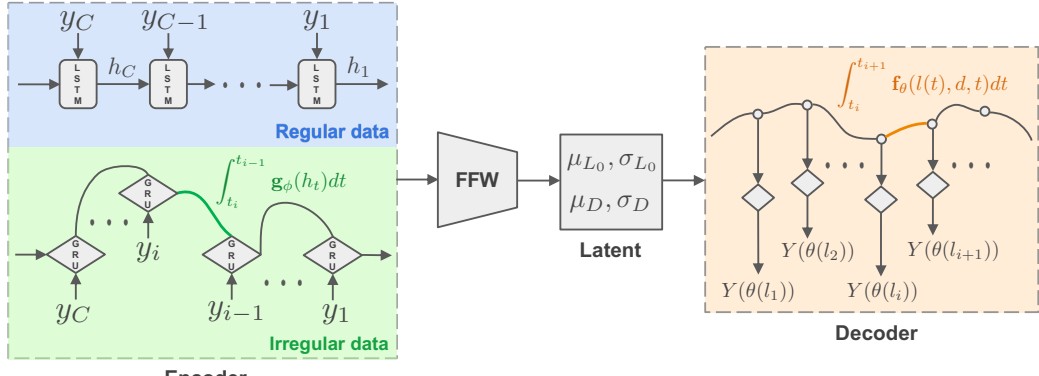

Figure 2: Illustration of the overall pipeline of the proposed SNODEP architecture.

pled, even though the underlying condition is the same. This further motivates us to employ a GRU-ODE encoder to capture the underlying dynamics and thus not be sensitive to irregularity.

## 3.2 STRUCTURED NEURAL ODE PROCESS (SNODEP)

**Encoder with Regularly Sampled Data.** To address the above issues, we propose a modified architecture where the encoder leverages Long Short-Term Memory (LSTM) (Hochreiter, 1997). The LSTM encoder is designed to capture dependencies between context points across time, allowing for a more informed and contextually-aware calculation of latent distributions $L_0(\theta_{l_0})$ and $D(\theta_d)$. We run the LSTM backward since we want the initial value of the latent variable $l_0$. Formally, the encoder takes the context sequence $\{(t_i, y_i)\}_{i \in \mathbb{I}_C}$ and computes hidden representations $\{h_i\}_{i \in \mathbb{I}_C}$:

$$h_i^{\mathrm{bwd}} = \mathrm{LSTM}_{\mathrm{bwd}}(y_i, h_{i+1}^{\mathrm{bwd}}), \ \text{ for } i \in \mathbb{I}_C.$$

**Encoder with Irregularly Sampled Data.** Recurrent networks assume inputs to be regularly spaced and have no consideration for the actual time the input was sampled, not applicable to irregularly sampled data (Rubanova et al., 2019). Thus, our hidden state varies according to a GRU-ODE:

$$\hat{h}_{i-1}^{\mathrm{bwd}} = h_i^{\mathrm{bwd}} + \int_{t_i}^{t_{i-1}} \mathbf{g}_\phi(h^{\mathrm{bwd}}(t)) \, dt, \ \ h_{i-1}^{\mathrm{bwd}} = \mathrm{GRU}(y_i, \hat{h}_{i-1}^{\mathrm{bwd}}), \ \text{ for } i \in \mathbb{I}_C,$$

where $\mathbf{g}_\phi$ is the network supposed to capture the time-dependent underlying dynamics of the hidden state, and GRU stands for the Gated Recurrent Unit (Cho, 2014), a gating mechanism typically employed in recurrent neural networks. For irregular data, our encoder uses the final hidden state $h_0^{\mathrm{bwd}}$ to calculate the parameters of the initial latent $l_0$ and control $d$, which then evolves to give us the time-varying probability distributions. But in Rubanova et al. (2019), the $h_0^{\mathrm{bwd}}$ is used to get the initial latent, $l_0$ which then evolves directly, giving us quantities of interest and there's no time-varying distribution involved. For both regular and irregular scenarios, the final hidden state from the backward pass gives us the representation $r = [h_0^{\mathrm{bwd}}]$, which is then used to parameterize the latent distributions $L_0$ and $D$, via a feed-forward layer (FFW in Figure 2).

**Latent distributions.** The latent distributions, $L_0(\theta_{l_0})$ and $D(\theta_d)$, are chosen based on the dataset. For gene-expression data, we model the latent distribution as a LogNormal distribution, to resemble the Poisson-like nature of the data:

$$l_0 \sim \mathrm{LogNormal}(\mu_{L_0}(r), \sigma_{L_0}(r)), \quad d \sim \mathrm{LogNormal}(\mu_D(r), \sigma_D(r)),$$

whereas for metabolic-flux and balance data, we use a Gaussian distribution:

$$l_0 \sim \mathcal{N}(\mu_{L_0}(r), \mathrm{diag}(\sigma_{L_0}(r))), \quad d \sim \mathcal{N}(\mu_D(r), \mathrm{diag}(\sigma_D(r))).$$

where $\mu_{L_0}, \sigma_{L_0}, \mu_D$ and $\sigma_D$ are learned functions. Using LogNormal ensures that we can resemble the Poisson-like nature of gene-expression data while still being able to use the re-parametrization trick (Kingma & Welling, 2013).

Table 1: Illustration of considered pathways with the number of genes, metabolites, and modules.

| Pathway | Num Genes | Num Metabolites | Num Modules |
|---|---|---|---|
| M171 | 623 | 70 | 168 |
| MHC-i | 281 | 6 | 9 |
| Iron Ions | 136 | 8 | 15 |
| Glucose-TCACycle | 84 | 11 | 15 |

**Decoder.** The decoder relies on evolving the latent variable $l(t)$ over time based on a neural ODE. For a given latent state at time $t_0$, the evolution is governed by:

$$l(t_i) = l_0 + \int_{t_0}^{t_i} \mathbf{f}_\theta(l(t), d, t) \, dt,$$

where $\mathbf{f}_\theta$ represents the dynamics defined by the Neural ODE, and $d$ is used for tuning the trajectory. At each target time $\{t_i\}_{i \in \mathbb{I}_\mathcal{T}}$, the latent state $l(t_i)$ is used to determine the target distributions. For gene expressions, we model the predicted distributions as a Poisson distribution:

$$y_i \sim \text{Poisson}(\lambda_y(l(t))) \quad \text{for } i \in \mathbb{I}_\mathcal{T}.$$

Whereas for metabolic flux and balances, we model the predicted distributions as a Gaussian:

$$y_i \sim \mathcal{N}(\mu_y(l(t)), \sigma_y(l(t))) \quad \text{for } i \in \mathbb{I}_\mathcal{T},$$

where $\lambda_y$, $\mu_y$ and $\sigma_y$ are again learned functions. The decoding distributions are meant to capture the nature of the corresponding data. The output distribution is motivated by the nature of distribution that we observe in the datasets as seen in Figure 9. During inference, we use the learned $\mathbf{f}_\theta$ to give latent values over unseen timesteps, from $\mathcal{V}$, as well.

### 3.3 OPTIMIZATION OBJECTIVE

Since the generative process is highly nonlinear, the true posterior is intractable. Thus, the model is trained using the amortized variational inference method using the evidence lower bound (ELBO):

$$\mathbb{E}_{q(l_0, d | \mathcal{T})} \left[ \sum_{i \in \mathbb{I}_\mathcal{T}} \log Y\left(y_i \mid l_0, d, t_i\right) + \log\left(\frac{L_0\left(l_0 \mid \mathcal{C}\right)}{L_0\left(l_0 \mid \mathcal{T}\right)}\right) + \log\left(\frac{D\left(d \mid \mathcal{C}\right)}{D\left(d \mid \mathcal{T}\right)}\right) \right], \tag{2}$$

where the expectation is taken over joint latent distribution $q(l_0, d) = L_0(\theta_{l_0}) \times D(\theta_d)$.

## 4 EXPERIMENTS

### 4.1 EXPERIMENTAL SETTINGS

**Datasets.** We use the gene-expression time-series dataset from Ori et al. (2021), which investigates the differentiation of human pluripotent stem cells into lung and hepatocyte progenitors using single-cell RNA sequencing to map the transcriptional changes during this process. The gene-count matrix has dimensions $10667 \times 26936$, with 10667 cells and 26936 genes. The gene expression is counted regularly across 16 days in batches with $\mathbb{B}_t$ being the index set of cells being counted on day $t$ and $|\mathbb{B}_0| + |\mathbb{B}_1| + \ldots + |\mathbb{B}_{15}| = 10667$. For each cell batch $\mathbb{B}_t$ and a given metabolic pathway, we only consider genes responsible for encoding the metabolites from the pathway. Table 1 summarizes the four metabolic pathways from Alghamdi et al. (2021) we considered in this study. Alghamdi et al. (2021) considered the metabolic reactions from the KEGG database (Kanehisa & Goto, 2000), import and export reactions, and reorganized them into modules based on the topological structure. This reorganization is, in essence, the simplification of the system of reactions by coercing connected reactions into a module. Thus, when we say flux or balance, we mean it with regard to a module.

**Methods.** We compare performances on model architectures, including neural process (NP) (Garnelo et al., 2018), neural ODE process (NODEP) (Norcliffe et al., 2021), and our structured neural

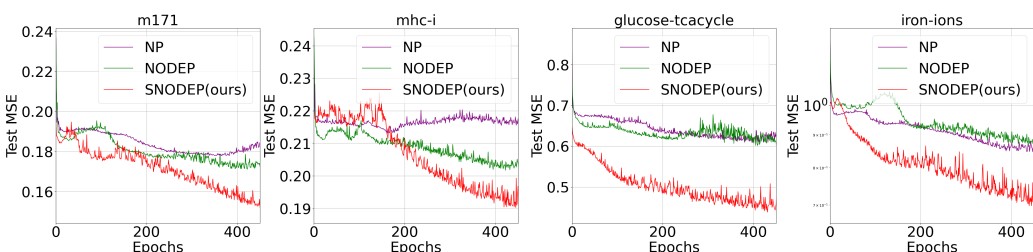

Figure 3: Comparison of test-MSE in log-scale between NP, NODEP, and SNODEP across different metabolic pathways on ground-truth gene-expression time-series data.

ODE process (SNODEP). We treat NP architecture as a baseline model to get insights on modeling our problems as a differentiable random process without considering the underlying dynamics. We also compare performances between NODEP and SNODEP, which have a neural-ODE decoder, with the latter exploiting sequential relationships between the context points via its encoder.

**Hyperparameters.** We vary the context length on the largest metabolic pathway, M171, to specify the hyperparameter setup for context length and train-test splits (see Appendix D). We observe that setting the context length as $8$ had a small test-MSE, corresponding to a $80/20$ split for train and test timesteps available. Thus for the experiments below, we set our context as $\mathbb{I}_{\mathcal{C}} = \{0, 1, \ldots, 8\}$ and our target as $\mathbb{I}_{\mathcal{T}} = \{0, 1, \cdots, 12\}$, while at inference, we predict for all the timesteps $\mathbb{I}_{\mathcal{V}} = \{0, 1, \ldots, 15\}$. Our training input is a sample $y \sim \Pi_{t=0}^{|\mathcal{T}|} Y(\theta_y(t))$ with context being $y[0 : |\mathcal{C}|]$ and target being $y[0 : |\mathcal{T}|]$. And during testing, we sample $y \sim \Pi_{t=0}^{|\mathcal{V}|} Y(\theta_y(t))$. For gene-knockout experiments, we set the train-test split of gene-knockout subsets as $80/20$.

**Evaluation Metric.** We use the MSE loss to measure model performance in predicting unseen timesteps of time-varying distributions. Let $s \sim Y_s$ and $s_* \sim Y_{s_*}$, where $Y_s$ is the learned distribution, and $Y_{s_*}$ is the ground-truth distribution, where $s, s_* \in \mathbb{R}$. Let $\mu_r$ and $\mathrm{Var}(r)$ be the mean and variance for any random variable $r$. Then the *Mean Squared Error* (MSE) is given by:

$$\mathbb{E}\left[(\mu_s - s_*)^2\right] = \mathbb{E}\left[\mu_s^2 + s_*^2 - 2\mu_s s_*\right] = \mathrm{Var}(s_*) + (\mu_s - \mu_{s_*})^2.$$

Assuming independence between dimensions, $\mathrm{MSE} = \sum_{i=1}^{d} \mathrm{Var}(s_*, i) + (\mu_{s,i} - \mu_{s_*,i})^2$ for $s \in \mathbb{R}^d$. For gene-expression data, assume $Y_s = \mathrm{Poisson}(\lambda)$ and $Y_{s_*} = \mathrm{Poisson}(\lambda_*)$. We thus get $\mathrm{MSE} = \sum_{i=1}^{d} \left[\lambda_{*,i} + (\lambda_i - \lambda_{*,i})^2\right]$. Note that gene-expression data is usually very sparse (Figures 9a and 9b), and hence $\lambda_g$ is usually very low. So in this case, minimizing MSE essentially boils down to getting as close to the Poisson approximation as possible. For metabolic flux and balance data, suppose $Y_s = \mathcal{N}(\mu, \sigma^2)$ and $Y_{s_*} = \mathcal{N}(\mu_*, \sigma_*^2)$. Then, we have $\mathrm{MSE} = \sum_{i=1}^{d} \left[\sigma_{*,i}^2 + (\mu_i - \mu_{*,i})^2\right]$. Since we observe the estimated flux and balance are of low variances (Figures 9c-9f), minimizing MSE essentially boils down to bringing the model mean $\mu$ closer to ground-truth mean $\mu_*$.

## 4.2 GENE-EXPRESSION

Ideally, we would like to collect the ground-truth metabolic flux and balance at an individual cell or tissue level. However, this is difficult because there is very little data on them. Gene-expression counts can be considered as a rough approximation for the concentration of proteins, metabolites, and enzymes they encode since they are highly correlated. Specifically, mRNA molecules are transcribed at a certain rate from the template DNA strand, which are then translated into proteins at some rate. Thus, we explore the timestep prediction task on log-normalized and scaled gene-expression time-series data. Here, $Y = G(\theta_g(t))$, which is defined in Section 2.

From Figure 3, we can clearly see that SNODEP achieves much lower MSE across different pathways, showing the efficacy of our proposed SNODEP. Both setting the sampling distribution as Poisson and using the contextual information for the latent variables, in conjunction, help in obtaining better performance. Even though we are working with ground-truth gene expressions, this result should encourage further study on ground-truth flux datasets.

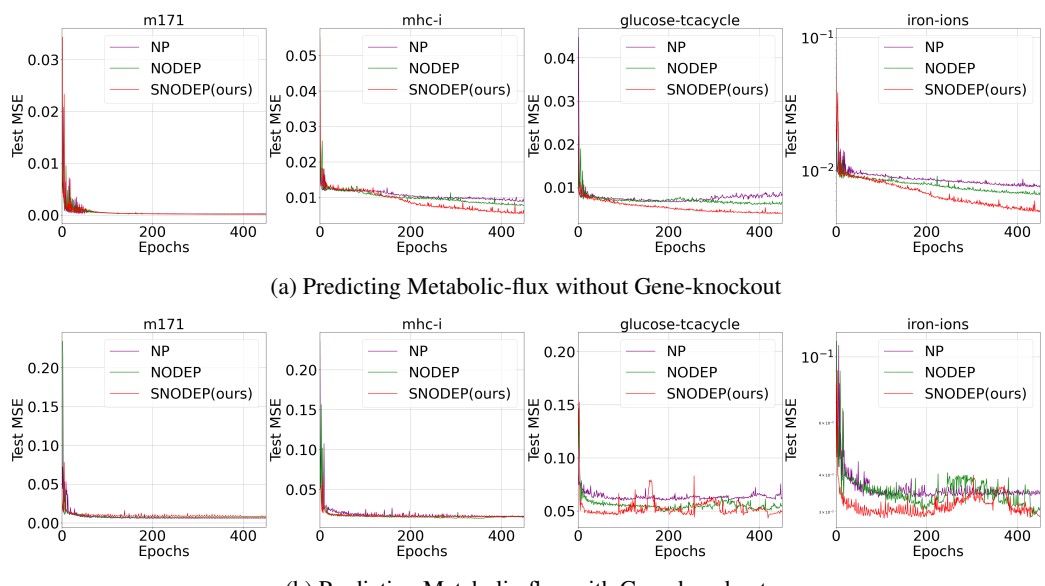

(a) Predicting Metabolic-flux without Gene-knockout

(b) Predicting Metabolic-flux with Gene-knockout

Figure 4: Comparison of test-MSE in log-scale between NP, NODEP, and SNODEP across different metabolic pathways on the scFEA-estimated metabolic-flux data with and without gene-knockout.

### 4.3 METABOLIC-FLUX

Applying techniques from single-cell flux estimation analysis on the gene-expression data, we obtain samples of metabolic fluxes for the metabolic pathways. Specifically for each time $t$, given a metabolic pathway with $d$ genes and $u$ modules with gene-expression matrix $\mathbf{G}_t \in \mathbb{R}^{d \times |\mathbb{B}_t|}$, we get $S_t^f(\mathbf{G}_t) = \mathbf{M}_t^f$, where $\mathbf{M}_t^f \in \mathbb{R}^{u \times |\mathbb{B}_t|}$ is the flux values for cell batch $\mathbb{B}_t$. From Figure 4a, we can observe that SNODEP performs generally better than the other two methods across different metabolic pathways, though the difference is not visually significant in some of them. We hypothesize that this is due to the nature of distributions as seen for some modules in Figures 9c and 9d, they have low variances, and if the mean of the distributions varies in an uncomplicated manner like linear or Markovian, incorporating the context in the latent is expected not to help much.

**Gene-knockout.** Gene-knockout experiments are meant to simulate the effect of disturbances in the pathway, such as the effect of any drug or environmental stress. Algorithm 1 in Appendix C describes the algorithmic form for our creation of knockout data. We model this by assuming that the gene expression level is correlated with how sensitive the metabolic pathway is with respect to the enzymes/proteins encoded by the gene. We consider $k$ most-expressed genes in the dataset and sample random subsets of these $k$ genes with the maximum cardinality of $k//2$. We call these random subsets as knockout sets where the gene expression for the genes contained is set to zero. We again calculate flux samples using scFEA (Appendix B.2) corresponding to each of knockout set, with train and test containing data corresponding to different knockout sets. In our experiments, we set $k = 20$ and the number of subsets $S = 5$ for all pathways. Figure 4b shows that our methodology is robust to gene knockout predictions, and overall, SNODEP performs better than NP and NODEP. This validates that we can use our model to predict behaviors of unseen gene knockout configurations experiments and unseen timesteps.

### 4.4 METABOLIC-BALANCE

Once we get the flux values for all the modules, we can immediately obtain the change in concentration of a particular metabolite, known as the balance in flux balance analysis, by multiplying the flux with the stoichiometric matrix. We thus perform analogous experiments as in Section 4.3, where for each time $t$ for a metabolic pathway with $d$ genes and $v$ metabolites with gene-expression matrix $\mathbf{G}_t \in \mathbb{R}^{d \times |\mathbb{B}_t|}$ we get $S_t^b(\mathbf{G}_t) = \mathbf{M}_t^b$, with $\mathbf{M}_t^b \in \mathbb{R}^{v \times |\mathbb{B}_t|}$ as defined in Section 2. Figure 5a

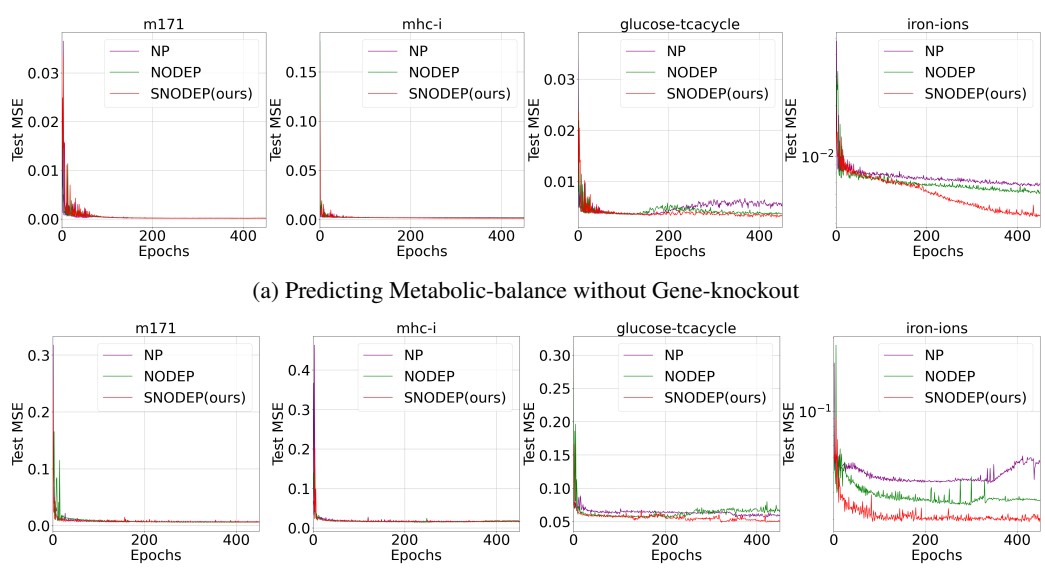

(a) Predicting Metabolic-balance without Gene-knockout

(b) Predicting Metabolic-balance with Gene-knockout

Figure 5: Comparison of test-MSE in log-scale between NODEP and SNODEP across metabolic pathways on scFEA-estimated metabolic-balance data with and without gene-knockout.

shows SNODEP generally outperforms NODEP, especially for the Iron Ions pathway. We believe that the performance is similar in pathways like MHC-i and M171 due to the simplistic nature of distributions (Figures 9e and 9f), akin to what we have mentioned in Section 4.3. Similar to the previous section, we follow the steps mentioned in Algorithm 1 to get the metabolic balance samples corresponding to gene knockout, and the test MSE is shown in Figure 5b. We can observe that the overall performance of SNODEP is better than that of NP and NODEP for all pathways.

## 4.5 IRREGULARLY SAMPLED TIMESTEPS

Data collection in experiments involving temporal profiling of gene expression is often performed irregularly (Rade et al., 2023; Nouri et al., 2023). Therefore, we also performed experiments where the points are irregularly sampled. To tackle the irregularity, we use GRU-ODE (Rubanova et al., 2019) to calculate latent distributions 3. Our context $\mathbb{I}_{\mathcal{C}}$ and target $\mathbb{I}_{\mathcal{T}}$ are similarly chosen to earlier sections, and we randomly set data from a fraction of timesteps to zero for each batch. During test time, we predict the remaining unseen timesteps. Figure 6 depicts heatmap visualizations of the difference between MSE of NODEP and SNODEP with GRU-ODE encoder. Entries in the heatmap with a positive value indicate that our SNODEP outperforms NODEP, and the higher the value is, the better the performance is. The negative values, where NODEP has a smaller test-MSE, are very low. We clearly see that SNODEP outperforms NODEP most of the time, especially towards lower frequencies, confirming the value of our model on irregularly sampled data.

## 5 CONCLUSION AND FUTURE WORK

In this work, we have shown how to get the time-varying metabolic flux of a system using genomics data rather than metabolomics data, which is much harder to procure. Through our framework, we intend to use the learned dynamics to generate quantities from future time steps and unseen gene-knockout configurations without any particular domain expertise. Nevertheless, we want to point out that our results with respect to flux and balance and their corresponding gene-knockout results are on data estimated via scFEA. Ideally, we would've preferred a gene-expression time-series that was sampled keeping metabolic pathways in mind, meaning time-series for normal conditions and several metabolic stresses, along with ground-truth metabolic flux and balance measurement. Such an experiment should also have the alternative DFBA formulation available so that we can benchmark

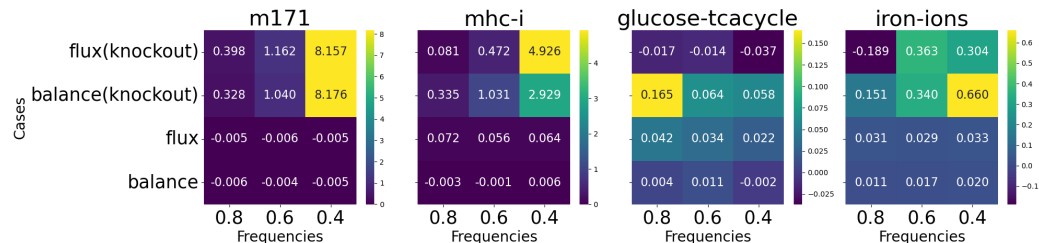

Figure 6: Heatmap of test-MSE difference ($\times 10^{-2}$) between NODEP and SNODEP with GRU-ODE encoder across metabolic pathways for flux, balance, and their knockout versions. Frequency refers to the fraction of the timesteps present. In Appendix F, we provide the corresponding tables.

our method with it. However, we could not find such an open-sourced dataset, so we provided our evaluations on scFEA estimated values instead of an ideal real-world dataset. Apart from such an evaluation, several future directions could be taken, like making the scFEA methods differentiable, enabling a single end-to-end differentiable pipeline, incorporating hypergraph structure into them, modifying the loss and distribution appropriately for the sparsity of gene-expression data, and exploring non-parametric probability estimations for the decoder, to name a few. We believe our work can also be helpful for integrating genomic and metabolomic data by using our pre-trained framework to fine-tune metabolomic data, for example. In conclusion, we believe our work can serve as a starting point for several interesting directions in making metabolic analysis more scalable.

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

# A  TERMINOLOGY

The key notations and terms used throughout this paper are defined below for clarity.

## A.1  INDICES AND SETS

- $t \in \{t_1, t_2, \ldots, t_V\}$: Discrete timesteps where measurements are collected, with $V$ representing the total number of timesteps.

- $\mathbb{B}_t$: The set of cells whose gene counts are measured at timestep $t$, such that the total number of cells across all timesteps is $N = \sum_t |\mathbb{B}_t|$.

- $\mathbb{I}_{\mathcal{C}} = \{1, \ldots, C\}$: Index set representing context points (i.e., the known timesteps used during model training).

- $\mathbb{I}_{\mathcal{T}} = \{1, \ldots, T\}$: Index set representing target points (i.e., the timesteps over which the model makes predictions).

- $\mathcal{V}$: The set of including unseen timesteps for which we aim to make predictions.

## A.2  GENE EXPRESSION DATA

- $K$: Total number of genes measured.

- $N$: Total number of cells.

- $g_{i,t} \in \mathbb{R}^d$: Gene-expression vector for cell $i$ at time $t$, with $d$ representing the number of genes relevant to a particular metabolic pathway.

- $\mathbf{G}_t \in \mathbb{R}^{d \times |\mathbb{B}_t|}$: Gene-expression matrix at time $t$ for the set of cells $\mathbb{B}_t$.

## A.3  METABOLIC QUANTITIES

- $m_{i,t}^f \in \mathbb{R}^u$: Metabolic-flux vector for cell $i$ at time $t$, with $u$ representing the number of modules.

- $m_{i,t}^b \in \mathbb{R}^v$: Metabolic-balance vector for cell $i$ at time $t$, with $v$ representing the number of metabolites.

- $\mathbf{M}_t^f \in \mathbb{R}^{u \times |\mathbb{B}_t|}$: Matrix of metabolic-fluxes for the set of cells $\mathbb{B}_t$ at time $t$.

- $\mathbf{M}_t^b \in \mathbb{R}^{v \times |\mathbb{B}_t|}$: Matrix of metabolic-balances for the set of cells $\mathbb{B}_t$ at time $t$.

## A.4  GENE-KNOCKOUT QUANTITIES

- $S$: The subset of gene-knockout configurations sampled.

- $k$: The number of most expressed genes selected for knockout.

- $\tilde{g}_{i,t,s} \in \mathbb{R}^d$: Gene expression vector for cell $i$ at time $t$ under the $s$-th knockout configuration (where $s \in \{1, \ldots, S\}$).

- $\tilde{\mathbf{G}}_{t,s} \in \mathbb{R}^{d \times |\mathbb{B}_t|}$: Gene-expression matrix at time $t$ for the set of cells $\mathbb{B}_t$ under the $s$-th knockout configuration.

- $\tilde{m}_{i,t,s}^f \in \mathbb{R}^u$: Metabolic-flux vector for cell $i$ at time $t$ under the $s$-th knockout configuration.

- $\tilde{m}_{i,t,s}^b \in \mathbb{R}^v$: Metabolic-balance vector for cell $i$ at time $t$ under the $s$-th knockout configuration.

- $\tilde{\mathbf{M}}_{s,t}^f \in \mathbb{R}^{u \times |\mathbb{B}_{s,t}|}$: Matrix of metabolic-fluxes for the set of cells $\mathbb{B}_t$ at time $t$ under the $s$-th knockout configuration.

- $\tilde{\mathbf{M}}_{s,t}^b \in \mathbb{R}^{v \times |\mathbb{B}_{s,t}|}$: Matrix of metabolic-balances for the set of cells $\mathbb{B}_t$ at time $t$ under the $s$-th knockout configuration.

### A.5 LATENT VARIABLES AND NEURAL ODE PROCESS

- $L_0(\theta_{l_0})$: Latent distribution of the initial state of the hidden representation $l_0$ for the cells, parameterized by $\theta_{l_0}$.
- $D(\theta_d)$: Latent distribution of an auxiliary variable $d$, parameterized by $\theta_d$, used to control the trajectory of the latent variables.
- $l(t_i) \in \mathbb{R}^z$: Latent state at time $t_i$ evolved from $l_0$ over time, where $z$ denotes the latent dimension.

### A.6 NEURAL NETWORKS AND FUNCTIONS

- $\mathbf{f}_\theta$: Neural network modeling the evolution of the latent state $l(t)$ through a Neural ODE.
- $\mathbf{g}_\phi$: Neural network modeling the evolution of the hidden state in the irregularly sampled case.
- $\mu_{L_0}(r), \sigma_{L_0}(r)$: Mean and standard deviation functions for the latent distribution $L_0$, parameterized by the hidden representation $r$.
- $\mu_D(r), \sigma_D(r)$: Mean and standard deviation functions for the latent distribution $D$.
- $\mu_y(l(t)), \sigma_y(l(t))$: Functions parameterizing the distribution of target outputs (gene expression, metabolic flux, or balance) at time $t$, based on the evolved latent state $l(t)$.
- $\lambda_y(l(t))$: Rate parameter for the Poisson distribution used to model gene-expression data.

### A.7 DISTRIBUTIONS AND LOSS FUNCTION

- $G(\theta_g(t))$: Distribution of gene expression at time $t$, parameterized by $\theta_g(t)$.
- $M^f(\theta_f(t))$: Distribution of flux at time $t$, parameterized by $\theta_f(t)$.
- $M^b(\theta_b(t))$: Distribution of balance at time $t$, parameterized by $\theta_b(t)$.
- $\tilde{M}^f(\theta_f(t))$: Distribution of flux under gene-knockout at time $t$, parameterized by $\theta_f(t)$.
- $\tilde{M}^b(\theta_b(t))$: Distribution of balance under gene-knockout at time $t$, parameterized by $\theta_b(t)$.
- $Y(\theta_t)$: General distribution (i.e., gene expression, flux and balance, and their gene-knockout variations) at time $t$, parameterized by $\theta_t$.
- ELBO: Evidence lower bound, the objective function used for model optimization, combining the log-likelihood of observed data and the Kullback-Leibler (KL) divergence between the true and approximate posterior distributions.

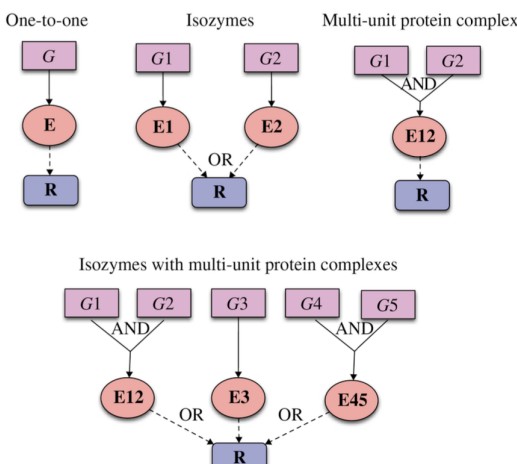

Figure 7: Different types of GPR relationships. G, E and R denote genes, enzymes and reactions, respectively. Solid arrows are enzyme production, while dashed arrows denote catalyzing a reaction.

## B  BACKGROUND

### B.1  FLUX BALANCE ANALYSIS

Flux balance analysis (FBA) is by far the most widely used computational method for analyzing stoichiometric-based genome-scale metabolic models. In this section, we introduce the optimization formulation of FBA and the meaning of the terms in the formulation. We refer readers to Maranas & Zomorrodi (2016) for a more detailed description.

**Static FBA.** In its most general form, FBA is formulated as an LP problem (Dantzig, 2002) maximizing or minimizing a linear combination of reaction fluxes subject to the conservation of mass, thermodynamic, and capacity constraints. The most widely used objective function in the FBA of metabolic networks is the maximization of the biomass reaction flux built upon the assumption that a cell is striving to maximally allocate all its available resources towards growth or maximizing the biomass, which is a predefined reaction that includes all the components required for cell growth like amino acids, nucleotides, lipids, and cofactors, etc. The formulation is as follows:

$$
\begin{aligned}
\text{maximize} \quad & z = \sum_{j \in \{\text{biomass}\}} c_j v_j \\
\text{subject to} \quad & \sum_{j \in J} S_{ij} v_j = 0, \quad \forall i \in I \\
& \text{LB}_j \leq v_j \leq \text{UB}_j, \quad \forall j \in J.
\end{aligned}
\tag{3}
$$

Equation 3 assumes that the time constants for metabolic reactions are very small, hence a pseudo steady state constraint $\boldsymbol{S} \cdot \boldsymbol{v} = 0$. Here, $z$ is the combination of reaction fluxes involved in the biomass, $v_j$ is the $j$-th reaction flux, $I$ is the set of all reactants, $J$ is the set of the reactions, $S_{ij}$ is the stoichiometric coefficient of reactant $i$ in reaction $j$. $\text{LB}_j$ and $\text{UB}_j$ are the lower and upper bounds on the rate of flux for reaction $j$, which depend on several factors such as reversibility based on Gibbs Free Energy ($\Delta G$), type of reaction, etc. These constraint values typically are hard to determine since they need to be meticulously determined experimentally for each case.

Depending on the use case, the constraints and the objective change. For example, in the case of simulating gene knockouts, we set $v_j = 0, \forall j \in J^{\text{KO}}$, and gauge the gene essentiality via the GPR relationships (see Figure 7), by targeting the biomass to be less than a threshold (no cell growth) when we want to kill cells, e.g., cancer cells. In metabolic engineering, we want overproduction of a target metabolic so our objective changes accordingly, and we can try different substrates, growth conditions, and gene knockout/knockin combinations towards overproduction.

**Dynamic FBA.** There are, however, processes where the time constants of the reactions are higher, like in the case of transcriptional regulation (minutes) or cellular growth (several minutes or hours) (Maranas & Zomorrodi, 2016). In such cases, the steady-state assumption $\boldsymbol{S} \cdot \boldsymbol{v} = 0$ isn't valid. Mahadevan et al. (2002) forego this steady-state assumption and study dynamic flux balance analysis in the context of Diauxic growth in E. Coli. Their study explores two separate formulations of dynamic FBA, namely (a) *Dynamic Optimization-based DFBA* and (b) *Static Optimization-based DFBA*. In (a), the optimization objective is integrated over the entire time duration via a dynamic function, while in (b), the batch is divided into intervals, and the LP is solved at the starting timestep of each interval. More precisely, (b) has the following formulation:

$$
\begin{aligned}
\text{maximize} \quad & z(t) = \sum_{j \in \{\text{biomass}\}} c_j v_j(t) \\
\text{subject to} \quad & x_i(t + \Delta T) \geq 0, \quad \forall i \in I \\
& v_j(t) \geq 0, \quad \forall j \in J \\
& \hat{\boldsymbol{c}}(\boldsymbol{z}(t))\boldsymbol{v}(t) \leq 0, \quad \forall t \in [t_0, t_f] \\
& |v_i(t) - v_i(t - \Delta T)| \leq \dot{v}_{i\max}\Delta T, \quad \forall t \in [t_0, t_f], \quad \forall i \in I \\
& x_i(t + \Delta T) = x_i(t) + \sum_{j \in J} S_{ij} v_j \Delta T, \quad \forall i \in I.
\end{aligned}
\tag{4}
$$

Here, $z$ is the combination of reaction fluxes involved in the biomass, $x_i$ is $i$-th reactant balance, $v_j$ is $j$-th reaction flux, $I$ is the set of all reactants, $J$ is the set of the reactions, and $S_{ij}$ is the stoichiometric coefficient of reactant $i$ in reaction $j$. In addition, $\hat{c}$ is a function representing nonlinear constraints that could arise due to consideration of kinetic expressions for fluxes, and $t_0$ and $t_f$ denote the initial and the final timestamps, respectively.

The static-DFBA formulation has much fewer parameters to solve for and is, therefore, more scalable. We would like to highlight the fact that static DFBA can be treated as a series of static FBAs that are solved locally for each timestep. In our work, for estimating metabolic flux from gene expression using techniques from Alghamdi et al. (2021), we thus solve for flux values for each timestep at the beginning of an interval.

### B.2 SINGLE-CELL FLUX ESTIMATION ANALYSIS

Singe-cell flux estimation analysis (scFEA) (Alghamdi et al., 2021) is a computational method to infer single-cell fluxome from single-cell RNA-sequencing (scRNA-seq) data. And we use it to estimate metabolic flux for the gene-expression data considered in our study. In scFEA, they reorganize the metabolic maps extracted from the KEGG database (Kanehisa & Goto, 2000), transporter classification database (Saier et al., 2006), biosynthesis pathways, etc., into factor graphs of metabolic modules and metabolites. We use the provided genes for metabolic pathway mappings in scFEA for several organisms to estimate metabolic fluxes.

Flux estimation is a neural network-based optimization problem where the likelihood of tissue-level flux is minimized. In particular, the network iteratively minimizes the flux balance, $\mathcal{L}_k^*$ with respect to each intermediate metabolite $C_k$:

$$
\mathcal{L}_k^* = \sum_{j=1}^{N} \left( \sum_{m \in \mathbb{F}_{\text{in}}^{C_k}} \text{Flux}_m^{(j)} - \sum_{m' \in \mathbb{F}_{\text{out}}^{C_k}} \text{Flux}_{m'}^{(j)} \right)^2
$$
$$
+ \sum_{k'} W_{k'} \sum_{j=1}^{N} \left( \sum_{m \in \mathbb{F}_{\text{in}}^{C_{k'}}} \text{Flux}_m^{(j)} - \sum_{m' \in \mathbb{F}_{\text{out}}^{C_{k'}}} \text{Flux}_{m'}^{(j)} \right)^2.
\tag{5}
$$

Let $\mathbb{G}_j^m = \{G_{i_1, j}^m, \ldots, G_{i_m, j}^m\}$ be the set of genes associated with metabolic module $F_m$, then here $\text{Flux}_m^{(j)} = f_{\text{nn}}^m(\mathbb{G}_j^m | \theta_m)$, flux of $F_m$ for $j$-th cell, is modeled as a multi-layer fully connected neural network with input $\mathbb{G}_j^m$ and $\theta_m$ being the parameters. Here, $C_{k'}$ denotes the Hop-2 neighbors of $C_k$ in the factor graph. $\mathbb{F}_{\text{in}}^{C_k}$ and $\mathbb{F}_{\text{out}}^{C_k}$ are the set of modules involved in production and consumption of $C_k$ respectively. The optimization problem of scFEA can be thought of as finding the optimal neural network configuration that gives us reaction fluxes from gene expressions, such that the total flux regarding metabolites is minimized when considered across all tissues.

## C   GENE KNOCKOUT ALGORITHM

---

**Algorithm 1** Gene-knockout Flux and Balance Calculation

---

**Require:** Gene-expression dataset $\mathbf{G}_t$ for time $t$ for a pathway with genes $\mathbb{H} = \{g_1, g_2, \ldots, g_d\}$,
number of most expressed genes $k$, number of subsets $S$
1: Identify the top $k$ most expressed genes: $\mathbb{H}_k = \{g_{i_1}, g_{i_2}, \ldots, g_{i_k}\}$
2: **for** $j = 1, \ldots, S$ **do**
3:     Randomly sample subset $\mathbb{S}_s \subseteq \mathbb{H}_k$ where $|\mathbb{S}_s| \leq k/2$
4:     Set $g_i = 0$ for all $g_i \in \mathbb{S}_s$ and let the new dataset be $\tilde{\mathbf{G}}_t$
5:     Calculate flux samples $\tilde{\mathbf{M}}_t^f = S_t^f(\tilde{\mathbf{G}}_t)$ and balance samples $\tilde{\mathbf{M}}_t^b = S_t^b(\tilde{\mathbf{G}}_t)$ using the
scFEA method described in Appendix B.2 for each knockout set $\mathbb{S}_s$
6:     Add the knockout information to samples $\tilde{m}^f = [\tilde{m}^f, b^g]$ and $\tilde{m}^b = [\tilde{m}^b, b^g]$ where $b^g$ is
a binary array such that:

$$b_i^g = \begin{cases} 0 & \text{if } g_i \text{ is in the knockout set } \mathbb{S}_s, \\ 1 & \text{otherwise.} \end{cases}$$

7: **end for**
8: Divide $\{\{\tilde{m}_{i,t}^f\}_{s,i\in\mathbb{B}_t}, \{\tilde{m}_{i,t}^b\}_{s,i\in\mathbb{B}_t}\}_{s\in\{1\cdots S\}}$ into train and test sets with different knockout sets

---

## D   EFFECT OF VARYING CONTEXT LENGTH

To perform our experiments in Section 4, we need to know how many context-target timesteps our model needs to be able to predict the remaining time steps properly. For this, we used the gene-expression data of the M171 pathway since it is the largest, and for a context length $C$, our extra target length is $C//2$. In Figure 8, we see that our model is able to learn optimally after $C = 6$ context steps or $C + C//2 = 9$ total steps. With little context, the model essentially learns next-step prediction (e.g., for $C = 2$ or $3, C//2 = 1$), which does not perform well. This experiment thus validates that with enough context, our model can learn the latent dynamics and can be used to generate data from future unseen timesteps.

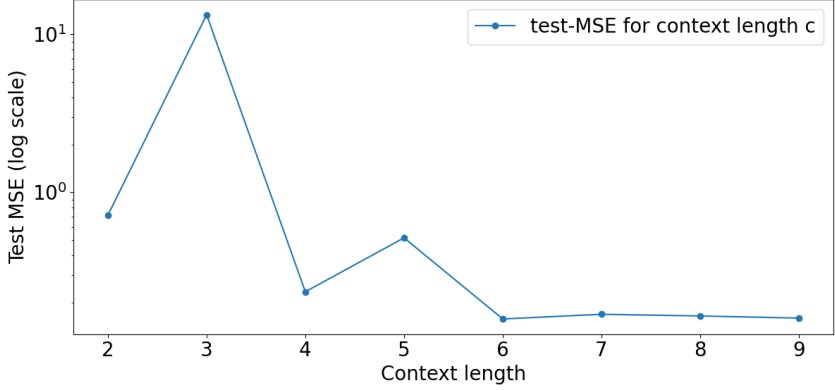

Figure 8: Curve plot of final test-MSE (in log scale) vs. context length.

# E  DATASET VISUALIZATION

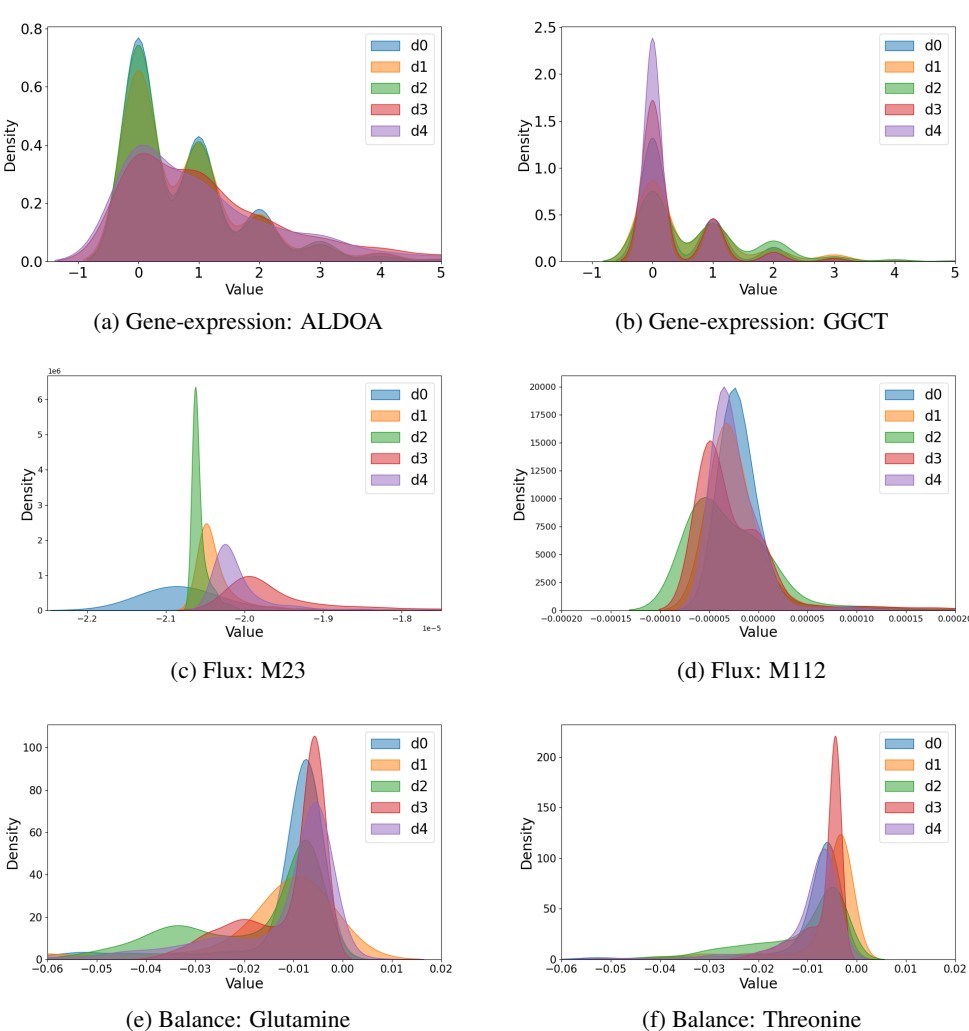

(a) Gene-expression: ALDOA

(b) Gene-expression: GGCT

(c) Flux: M23

(d) Flux: M112

(e) Balance: Glutamine

(f) Balance: Threonine

Figure 9: Kernel density estimate plots for the time-varying distributions of some gene, metabolite balance, and module flux from the largest metabolic pathway M171 for the first five days.

# F   IRREGULARLY SAMPLED DATA

We provide the detailed results in table format of the heatmap visualization for our experiments with irregularly sampled data (Figure 6) For each setting, the lowest value is highlighted in bold.

Table 2: Test-MSE ($\times 10^{-2}$) across different cases and frequencies for M171 pathway.

| Case | Frequency | NODEP | SNODEP (GRU-ODE) | Diff |
|---|---|---|---|---|
| flux (knockout) | 0.8 | 0.9434 | **0.5451** | 0.3983 |
| flux (knockout) | 0.6 | 1.6365 | **0.4743** | 1.1622 |
| flux (knockout) | 0.4 | 8.5318 | **0.3745** | 8.1573 |
| balance (knockout) | 0.8 | 0.9452 | **0.6174** | 0.3278 |
| balance (knockout) | 0.6 | 1.5794 | **0.5397** | 1.0397 |
| balance (knockout) | 0.4 | 8.5747 | **0.3985** | 8.1762 |
| flux | 0.8 | **0.0232** | 0.0285 | -0.0053 |
| flux | 0.6 | **0.0185** | 0.0245 | -0.0060 |
| flux | 0.4 | **0.0138** | 0.0188 | -0.0050 |
| balance | 0.8 | **0.0196** | 0.0255 | -0.0059 |
| balance | 0.6 | **0.0163** | 0.0202 | -0.0039 |
| balance | 0.4 | **0.0113** | 0.0160 | -0.0047 |

Table 3: Test-MSE ($\times 10^{-2}$) across different cases and frequencies for MHC-i pathway.

| Case | Frequency | NODEP | SNODEP (GRU-ODE) | Diff |
|---|---|---|---|---|
| flux (knockout) | 0.8 | 1.5262 | **1.4450** | 0.0812 |
| flux (knockout) | 0.6 | 1.7510 | **1.2789** | 0.4721 |
| flux (knockout) | 0.4 | 5.8514 | **0.9251** | 4.9263 |
| balance (knockout) | 0.8 | 1.7747 | **1.4393** | 0.3354 |
| balance (knockout) | 0.6 | 2.2862 | **1.2553** | 1.0309 |
| balance (knockout) | 0.4 | 3.8700 | **0.9410** | 2.9290 |
| flux | 0.8 | 1.0332 | **0.9609** | 0.0723 |
| flux | 0.6 | 0.8438 | **0.7881** | 0.0557 |
| flux | 0.4 | 0.5904 | **0.5262** | 0.0642 |
| balance | 0.8 | **0.1697** | 0.1722 | -0.0025 |
| balance | 0.6 | **0.1484** | 0.1494 | -0.0010 |
| balance | 0.4 | 0.1179 | **0.1123** | 0.0056 |

Table 4: Test-MSE ($\times 10^{-2}$) across different cases and frequencies for Iron Ions pathway.

| Case | Frequency | NODEP | SNODEP (GRU-ODE) | Diff |
|------|-----------|-------|------------------|------|
| flux (knockout) | 0.8 | **3.0829** | 3.2716 | -0.1887 |
| flux (knockout) | 0.6 | 3.0149 | **2.6521** | 0.3628 |
| flux (knockout) | 0.4 | 2.2513 | **1.9470** | 0.3043 |
| balance (knockout) | 0.8 | 3.3664 | **3.2149** | 0.1515 |
| balance (knockout) | 0.6 | 3.0049 | **2.6651** | 0.3398 |
| balance (knockout) | 0.4 | 2.6629 | **2.0024** | 0.6605 |
| flux | 0.8 | 0.8134 | **0.7822** | 0.0312 |
| flux | 0.6 | 0.6596 | **0.6310** | 0.0286 |
| flux | 0.4 | 0.4470 | **0.4137** | 0.0333 |
| balance | 0.8 | 0.6903 | **0.6788** | 0.0115 |
| balance | 0.6 | 0.5861 | **0.5693** | 0.0168 |
| balance | 0.4 | 0.4098 | **0.3900** | 0.0198 |

Table 5: Test-MSE ($\times 10^{-2}$) across different cases and frequencies for Glucose-TCACycle pathway.

| Case | Frequency | NODEP | SNODEP (GRU-ODE) | Diff |
|------|-----------|-------|------------------|------|
| flux (knockout) | 0.8 | **5.4157** | 5.4325 | -0.0168 |
| flux (knockout) | 0.6 | **4.6533** | 4.6676 | -0.0143 |
| flux (knockout) | 0.4 | **3.4707** | 3.5078 | -0.0371 |
| balance (knockout) | 0.8 | 5.6183 | **5.4531** | 0.1652 |
| balance (knockout) | 0.6 | 4.8713 | **4.8077** | 0.0636 |
| balance (knockout) | 0.4 | 3.5659 | **3.5082** | 0.0577 |
| flux | 0.8 | 0.6731 | **0.6308** | 0.0423 |
| flux | 0.6 | 0.5547 | **0.5210** | 0.0337 |
| flux | 0.4 | 0.3964 | **0.3741** | 0.0223 |
| balance | 0.8 | 0.3372 | **0.3329** | 0.0043 |
| balance | 0.6 | 0.2996 | **0.2881** | 0.0115 |
| balance | 0.4 | **0.2090** | 0.2113 | -0.0023 |

