# OpenReview forum: "Predicting Time-Varying Flux and Balance in Metabolic Systems using Structured Neural ODE Processes"
_ICLR.cc/2025/Conference — Submitted to ICLR 2025_

### Official Review · Reviewer_UXun · 2024-10-30

**Soundness:** 2
**Presentation:** 2
**Contribution:** 1
**Rating:** 3
**Confidence:** 2

**Summary:**

In this submission, the author propose a data-driven solution to Dynamic Flux Balance Analysis (DFBA), a classical tool from Systems Biology to analyze genome-scale metabolic network. At each timestep, DFBA solves a time-varying constrained optimization problem where the goal is to determine the value of the fluxes that lead to maximal biomass production, under time-varying biological constraints, which typically can only be brought by a  deep understanding of the system at hand.

Leveraging gene expression data in both control and knockout experiments thanks to single-cell flux estimation analysis, the authors aim at predicting time-resolved unseen gene expression, flux, balance, in both control and knockout experiments. This is achieved by means of Neural ODE processes (NODEP). Particularly, the main contribution of the paper is to cast DFBA in the framework of NODEP, and to highlight the need for a modification of NODEP to better handle the 1) irregularly sampled data and 2) non-gaussianity of certain distributions involved. This leads to Structural NODEP (SNODEP).

Finally, the authors carry a number of experiments on different metabolic pathways, demonstrating the clear improvement provided by SNODEP, particularly for sparsely sampled data across time.

**Strengths:**

- The paper is well-organized and well-motivated by a biological question.

- The empirical assessment clearly demonstrate the superiority of the proposed method.

**Weaknesses:**

- There is little to no technical novelty. The paper feels like the application of Neural ODE processes to DFBA, with minor tweaks to make it work better. Although the results are convincing, and there is definitely value in framing the DFBA problem in terms of Neural ODE processes, I am not sure that ICLR is a relevant venue for this work.

**Questions:**

None

---

> ### Author Response · Authors · 2024-11-26
> **Response to Reviewer UXun**
>
> We thank the reviewer for the provided insightful comments and detailed feedback. Please find our clarifications below:
>
> ---
>
> ### 1. Little to no technical novelty.
> The paper feels like the application of Neural ODE processes to DFBA, with minor tweaks to make it work better. Although the results are convincing, I am not sure that ICLR is a relevant venue for this work.
>
> We would like to emphasize why the considered problem is impactful. Metabolic flux measurements, particularly at the single-cell level, are both expensive and technically challenging. Determining the compounds in a solution requires mass or NMR spectroscopy, followed by isotope subtyping or similar techniques to measure reaction fluxes. These methods rely on highly specialized instruments and experimental setups, resulting in very limited ground-truth data. Machine learning and data-driven methods are therefore highly suited to address these challenges. Please refer to our General Response 1 for further details.
>
> While we acknowledge that several methodological components of our framework have appeared in earlier works individually and in different contexts, we argue that the combination of these components is novel in the context of our problem statement. Specifically, modeling flux-balance estimates as a random process with a biologically motivated approach is unique to our work. Please refer to our General Response 3 for additional context.

---

### Official Review · Reviewer_djzo · 2024-11-02

**Soundness:** 1
**Presentation:** 2
**Contribution:** 1
**Rating:** 3
**Confidence:** 4

**Summary:**

This papers proposes a new method (SNODEP) to predict temporal flux and balance in metabolic systems. The architecture relies on a Neural ODE approach where they model the joint temporal evolution of scRNAseq, flux, and balance. The model is then trained to predict the future of the trajectories based on a contextual window. The authors state that their machine learning contributions includes modifying the emission distribution, as well as a new structured encoder approach for encoding context. The experiment section shows comparison with neural processes and neural ode processes, and shows that SNODEP performs better on scRNAseq forecasting, as well as flux and balance prediction.

**Strengths:**

Predicting flux and balance from metabolic systems in a temporal way is a very interesting topic and the Neural ODE formalism seems well adapted for this task. Therefore, the motivations of the paper are clear and the intended impact is evident.

The results suggest than SNODEP outperforms the considered baselines, highlighting its potential for metabolic predictions.

**Weaknesses:**

A. The contributions of this paper are insufficient from a machine learning perspective.

1. changing the emission distribution of a neural ODE model is typically considered a design choice in all existing NODE architectures. Such that choosing a Poisson emission can hardly qualify as a contribution.
2. I am afraid that the identified limitation of the lack of structure for the conditioning is artificial. The authors state that : "The encoded representation is calculated using context points without any particular structure in NODEP. [...] The order between the context points and their sequential dependence on each other is not efficiently utilized." This may be the case for the specific NODEP model the authors refer too but this statement is inaccurate for most NODE models such as RNN-ODE (Rubanova et al.) or GRU-ODE (De Brouwer et al.)

Given that these are the two contributions laid out by the authors, the machine learning impact is minimal. That said, that does not take away anything from their impact for biological modeling (metabolic).

B. Modeling problems.

1. The authors appear to consider that cells taken at different time points form a time series. However, they recognize that "The challenge, however, lies in that gene expression trajectories for individual cells cannot be tracked over time since cells die once their gene expression is read.". I could not find anything in the present paper that directly addresses that challenge. Instead, it seems different cells at different time points are directly taken as a time series, which seems inaccurate. At the very least, the authors should make this assumption clear and motivate it.

C. Results.

1. Most of the results are given in the form of training curves which make rapid evaluation of the performance difficult. The scales of some subplot is sometimes off, such that the different methods cannot be distinguished.
2. I am pretty sure that RNN-ODE and GRU-ODE could also be used for this problem. I would encourage the authors to expand their list of baselines. I would also include very simple baselines like RNNs - or even one-step-ahead linear regressors.

**Questions:**

- Can you please address the concern regarding the destructive measurement process of scRNAseq ? It seems different cells at different time points are directly taken as a time series, which seems inaccurate.

- I didn't fully understand why the distribution of the latent is that important, given that you end up modulating it with a non-linear function for the final predictions. Could you please motivate this choice more rigorously ? Line 260 - Latent distributions.

---

> ### Author Response · Authors · 2024-11-26
> **Response to Reviewer djzo**
>
> We thank the reviewer for the provided insightful comments and detailed feedback. Please find our clarifications below:
>
> ---
>
> ### 1. Can you please address the concern regarding the destructive measurement process of scRNA-seq?
> It seems different cells at different time points are directly taken as a time series, which seems inaccurate. How do you address this challenge?
>
> We address this challenge by formulating our task as a *random process prediction* task. A random process can be interpreted as either a distribution over trajectories or a distribution over values at each time step; we adopt the latter perspective. As the reviewer correctly noted, sampling from prior time steps does not yield precise values for the next step, which is why we predict a distribution instead of a fixed value. This distribution inherently captures the underlying uncertainty in the prediction. Our approach models random processes in a structured manner, focusing on distributions rather than trajectories. Please refer to our General Response 2.
>
> ---
>
> ### 2. The lack of structure might be the case for NODEP but not for other NODE models such as RNN-ODE (Rubanova et al.) or GRU-ODE (De Brouwer et al.).
>
> While we agree that RNN-ODE and GRU-ODE introduce structure in their architectures, their outputs are trajectories rather than distributions. Our focus is on predicting distributions over time, not trajectories, as our problem is modeled as a random process. To the best of our knowledge, none of the existing neural process architectures incorporate structural modeling.
>
> ---
>
> ### 3. Most of the results are given in the form of training curves, which makes rapid evaluation of performance difficult.
> The scales of some subplots are sometimes off, making it hard to distinguish between methods.
>
> We chose to represent results as training curves to save space, given the comparison of three models across four metabolic pathways and five cases each. For irregularly sampled data, we have provided tables in the Appendix. Regarding the scale, we presented all plots on a log scale, but differences between metabolic pathways can lead to varying y-axis limits, which may appear inconsistent.
>
> ---
>
> ### 4. I am pretty sure that RNN-ODE and GRU-ODE could also be used for this problem.
> I encourage the authors to expand their list of baselines, including RNNs or even one-step-ahead linear regressors.
>
> While RNN-ODE and GRU-ODE could be adapted to this problem, these models predict values, not distributions. Estimating a distribution from these methods would require running each trajectory multiple times, which is computationally inefficient. Neural process models, by design, directly account for this need, which is why we focused on them instead. Please refer to our General Response 2.
>
> ---
>
> ### 5. I didn’t fully understand why the distribution of the latent is that important, given that you end up modulating it with a non-linear function for the final predictions. Could you please motivate this choice more rigorously?
>
> In our initial experiments, aligning the latent distribution with the final output distribution led to a slight reduction in MSE, which motivated this choice. However, as the reviewer correctly pointed out, the theoretical justification for this choice remains inconclusive. We acknowledge the need for a more exhaustive investigation and plan to explore this in future studies.

---

> > ### Comment · Reviewer_djzo · 2024-12-03
> > **Thank you for your response**
> >
> > I want to thank the authors for addressing my comments.
> >
> > Given your explanation about how you handle the lack of temporal follow-up of samples, it then seems you assume that starting from an initial condition at time step $t-1$, you reach any scRNA seq measurement at time step $t$. While this assumption can be somehow defended, it still is, in my understanding, an unusual assumption for scRNA data.
> >
> > RNN-ODE and GRU-ODE do model output gaussian distributions and not only point estimates. Using a Poisson emission distribution only changes the likelihood but does not require significant modeling changes.
> >
> > Despite all the merits of this work, I still believe the lack of technological novelties make it unfit for this conference at this stage, and I confirm my score.

---

### Official Review · Reviewer_J9xR · 2024-11-02

**Soundness:** 2
**Presentation:** 2
**Contribution:** 1
**Rating:** 3
**Confidence:** 3

**Summary:**

The paper proposes a forecasting architecture based on Neural Ordinary Different Equation Processes (NODEP) which they call Strutrude Neural ODE Process (SNODEP) and applies it to the problem of forecasting gene expression and metabolic flux and balance values across time. More specifically, given a time series of single-cell gene expression data, the authors first generate metabolic flux and balance values from this data using the previously published [scFEA algorithm](https://genome.cshlp.org/content/31/10/1867.short). They do this both for the actual data as well as for synthetic genetic knockout data produced by overwriting some genes' expression value to zero and then applying the scFEA algorithm to this semi-synthetic dataset. Given a partially observed time course of this data, the task then is to predict the value at future time points.

In order to solve the task, the authors modify the original NODEP architecture. NODEP parametrizes distributions over ODEs through an encoder-decoder framework. In this framework, the authors change the latent distribution from Gaussian to Log-normal and the sampling distribution/loss function from Gaussian to Poisson to better match the nature of the gene expression data. Moreover, they replace the order-invariant encoder architecture of the original NODEP framework with time-structured networks such as Long Short-Term Memory (LSTM) and Gated Recurrent Unit (GRU).

They apply this framework to one dataset of human pluripotent stem cell differentiation and observe improved MSE performance compared to Neural Process (NP) and NODEP baselines.

**Strengths:**

The specific proposed changes to the NODEP architecture look novel to me.

**Weaknesses:**

* **Limited overall contribution:** The paper makes two contributions, introducing the metabolic system forecasting problem as a test case of interest and developing a novel architecture to solve it. In the following, I will outline why I think it does not hit the mark in either of those areas.
* **Poorly motivated metabolic system forecasting problem**
    * While understanding metabolic flux and balances is useful, the authors don't make a strong point for why the forecasting problem is specifically impactful. They consider an 80%/20% data split, so if they could show near-perfect prediction of future values, this would correspond to a 20% saving in experimental budget. Is that impactful enough in this domain for people to adopt this? Moreover, they make no attempt in characterizing how good is "good enough". The only evaluation shown is mean squared error. Are there specific scientific questions that can be solved or cannot be solved at the attained accuracy as compared to the baselines?
    * ODE systems often come with the advantage of understanding inherent dynamics of the underlying system. It seems to me like the NODEP formulation foregoes this in favor of an abstract latent space. However, even latent spaces could lend themselves to interpretations.  Unfortunately, the authors do not show any interpretation of the learned systems.
* **Narrow scope of ML contributions**
    * While SNODEP definitely seems superior to NODEP in the presented experiments, the paper leaves me wondering which of the changes (different neural network architecture, different loss function, different latent distribution) is responsible for the performance boost. This could easily be explored in ablation studies.
    * To meet the bar of a significant machine learning contribution, I would have liked to see comparisons of SNODEP on problems in the original NODEP paper and related subsequent publications, not only one very bespoke biological problem setup.
* **Limited choice of baselines:** The considered baselines are verry narrow in scope, essentially copied over from the NODEP paper. If I understand the metabolic systems prediction task correctly as a time series prediction task, what about trying simpler methods (like LSTM or GRU directly applied to the data) or time series foundation models?
* **Poor presentation of problem & results:**
    * I could not fully understand the inputs and outputs to the method (see questions below).
    * I am confused as to why all the MSE results are shown as MSE curves over training epochs. This suggests no replicate experiments were performed and thus that no uncertainty quantification on the results can be performed. It would have been simpler to summarize the final MSE values as bar plots or tables with error bars/intervals.

**Questions:**

* I did not fully understand the inputs and outputs to the method. For example, are the gene expression values and metabolic flux and balance values treated as separate time series or does the method process them jointly?
* Could the problem be set up as a regular prediction task, with a simple LSTM or GRU architecture to predict, or are there specific problem characteristics that force the use of ODE based architectures?

---

> ### Author Response · Authors · 2024-11-26
> **Response to Reviewer J9xR**
>
> We thank the reviewer for the provided insightful comments and detailed feedback. Please find our clarifications below:
>
> ---
>
> ### 1. The authors don't make a strong point for why the forecasting problem is specifically impactful.
>
> The primary reason this problem is impactful is due to the high cost and difficulty of performing metabolic flux measurements, especially at the single-cell level. Determining flux requires specialized instruments and experimental setups, such as mass or NMR spectroscopy combined with isotope subtyping. As a result, ground truth data is scarce. Please refer to our General Response 1.
>
> The value of the performance from the 80/20 split isn't limited to the 20% savings in cost but rather demonstrates that our model has captured the underlying dynamics, providing confidence in using the predictions for downstream analyses. While we considered other splits, the limited time span of our data (16 days) restricts the number of viable splits, as some context time points are required for training. In the Appendix subsection *Effect of Varying Context Length*, we explored these variations and selected the 80/20 split accordingly.
>
> "How good is good enough?" is indeed an important question. Addressing it requires downstream analysis of predictions, such as comparing ground truth metabolic-flux/balance values or biomass objective values derived from predictions. As noted in the *Conclusions* section, this would require time-varying scRNA-seq data aligned with ground truth metabolic data. Unfortunately, such datasets are not publicly available due to the challenges of data collection, so we left this for future work. Meanwhile, we evaluate models based on MSE and prioritize those with lower values. Our model's ability to generalize the dynamics enables practitioners to perform gene-knockout studies or predict biomass yield without solving the dFBA optimization problem, which often requires domain expertise to formulate.
>
> ---
>
> ### 2. I did not fully understand the inputs and outputs of the method.
>
> We train separate models for each task: gene expression, metabolic flux, metabolic balance, metabolic flux with gene knockouts, and metabolic balance with gene knockouts. Each model adapts to regular or irregular sampling. During training, the encoder takes in context points, calculates the latent variables, samples the latent $l_0$ and $d$, and passes them to the decoder. The decoder predicts latent values for target points via a neural ODE, from which it derives distributions for target timepoints. During testing, we evaluate all available points, including unseen timepoints during training.
>
> ---
>
> ### 3. Could the problem be set up as a regular prediction task with simpler architectures?
>
> This is not a regular prediction task because we cannot track gene expression of individual cells over time; measurements destroy the cells. Consequently, flux and balance estimates cannot be determined for single cells across time, and the resulting trajectories are random processes with uncertainty about future values. Modeling this as a random process, which predicts a distribution instead of a point, is a natural fit.
>
> To achieve similar results with LSTMs or GRUs, multiple runs per trajectory would be required to estimate a distribution at each timestep, which is computationally inefficient. Furthermore, the chemical kinetics underlying these systems are commonly formulated as ODEs, motivating the use of neural-ODE models.
>
> ---
>
> ### 4. Baselines are from NODEP; what about trying simpler methods or time-series foundation models?
>
> Although simpler methods could be explored, this is not purely a time-series problem but a random process problem, where the predictions are distributions rather than points. Moreover, chemical kinetics are inherently described by ODEs, which aligns well with NODEP-based approaches. To estimate distributions using simpler time-series methods, multiple runs would be needed, making them less efficient. Please refer to our General Response 2.
>
> ---
>
> ### 5. Unfortunately, the authors do not show any interpretation of the learned systems.
>
> We chose a latent architecture for two reasons: (a) scalability, given the large number of genes/metabolites, and (b) suitability for irregularly sampled data, as shown in prior works like [1]. Since precise formulations of metabolic pathways via chemical kinetics are still ongoing, we did not focus on interpreting latent mappings but agree that exploring interpretability would be valuable.
>
> [1] Rubanova, Y., Chen, R. T. Q., & Duvenaud, D. (2019). Latent ODEs for Irregularly-Sampled Time Series.
>
> ---
>
> ### 6. Misc
> We appreciate the suggestion for ablation studies, and will incorporate in our revisions. We chose not to explore comparisons on other tasks because it diverges from our primary motivation of predicting metabolic flux and balance. However, we understand the importance of broader comparisons and will consider this in subsequent work.

---

> ### Comment · Reviewer_J9xR · 2024-11-26
> **Thank you for your response!**
>
> Thank you for your response, clarifying some aspects of the proposed method.
>
> > 1. Why is it impactful? Cost and complexity
>
> I understand that collecting the right type of data for the metabolic flux and balance prediction can be complex and costly. However, this limitation seems to be mostly addressed by scFEA in your current work, reducing the problem to collecting time-series scRNA-Seq data.
>
> > 3. Regular prediction task with simpler architectures. Distribution-level problem
>
> I agree that measurements are destructive so that you don't have paired data available. But if I understand correctly, you currently measure prediction quality exclusively at the level of the mean squared error on the mean of the distribution. I am fine with that, but as a prediction target, I still believe this would lend itself very nicely to a simple predictive architecture such as LSTM or GRU (ignoring the rest of the observed distribution).

---

> ### Author Response · Authors · 2024-11-26
> **Thanks for your further comments!**
>
> We thanks the reviewer for their further comments. Please find our responses below:
>
> ---
> ### 1.  This limitation seems to be mostly addressed by scFEA in your current work
>
> To an extent you're correct, however, in the real world settings, scRNA-seq data isn't usually considered for metabolic analysis, instead what we usually have are urine or blood samples from which we get the metabolomic profile. This is aligned with our methodology where our flux model is trained  *directly* on flux time samples.
>
> We use the scFEA framework to get *estimated* flux and balance data because the real world data is difficult to procure, and thus in our context the scFEA should be considered as a proxy for the real world flux-balance data-- something we *assume* in the section **Problem Formulation**.
>
> Although, we did perform experiments with single-cell gene-expression data for precisely for the reason that it's a real world data and is highly correlated with the amount of metabolic pathway molecules-- discussed in the section **4.2 Gene-expression**.
>
> Ideally, we would've preferred working directly with real world flux-balance data. However we were unable to find such open sourced datasets, and as a result we resorted to scFEA estimates. We discuss this in more detail in the section **Conclusions**, along with open questions. And we're trying our level best to procure real world data but till then, we think it's best to use scFEA as a proxy for real world data to make methodological progress.
>
> ---
> ### 2. I still believe this would lend itself very nicely to a simple predictive architecture such as LSTM or GRU (ignoring the rest of the observed distribution).
>
> We appreciate the suggestion and agree that LSTM/GRU-based approaches could serve as a valuable baseline. However, adapting these methods for our specific setting would require additional effort beyond a straightforward prediction task using LSTMs or GRUs, as these methods do not inherently consider the underlying dynamics or provide distributional outputs. That said, we do recognize the potential value in exploring this direction and will seriously consider it for future work. Thank you for highlighting this opportunity!

---

> ### Comment · Reviewer_J9xR · 2024-11-26
> **Thank you for your response!**
>
> I would like to thank the authors again for the clarifications provided. Sadly, overall, I maintain my evaluation. To change my score, the task itself would have to better motivated and explored in-depth to be convincing, and the technical novelty seems limited. In particular, the baseline suggestions raised by me and other reviewers remain unaddressed in the current version of the paper.

---

### Official Review · Reviewer_Vvou · 2024-11-04

**Soundness:** 2
**Presentation:** 2
**Contribution:** 2
**Rating:** 5
**Confidence:** 3

**Summary:**

This paper addresses the challenging problem of predicting metabolic flux and metabolic balance from single-cell data. The authors formulate this prediction problem as learning the underlying dynamics of metabolic flux and metabolic balance and propose a novel method, structured neural ODE process (SNODEP), based on neural ordinary differential equations (Neural ODEs). Through extensive empirical experiments, the authors demonstrate that their novel method, SNODEP, yields improved performance on metabolic flux and metabolic balance prediction relative to baseline methods.

**Strengths:**

The proposed method, SNODEP, presents several key contributions that help address the challenging problem of metabolic flux and balance modelling and prediction:

- SNODEP is able to effectively and efficiently model the dynamics of metabolic flux and balance, normally a challenging endeavour, by formulating metabolic flux and balance prediction as a data-driven problem.
- SNODEP is evaluated over an extensive set of different pathways and experiments to showcase its utility for predicting metabolic flux and balance.
- The proposed method can learn the dynamics of metabolic flux and metabolic balance in regularly and irregularly sampled time-series settings as well as under various gene-knockout conditions.

**Weaknesses:**

Although this work presents a novel perspective for tackling the problem of predicting metabolic flux and metabolic balance under a data-driven framework, it contains several unaddressed items that limit the overall quality and contribution of this work. For example:

- At times, the presentation and exposition of text in the manuscript is difficult to follow. For example, the learning objective in the problem formulation section, which appears to be a key section of the paper, is not very clear. Moreover, equations throughout the text don't have labels -- it would be helpful to numerically label all equations and reference them in the text. See the questions below for further details.
- Evaluation metrics are limited, only the mean squared error (MSE) metrics is considered. This provides a narrow view and comparison of model performance relative to baselines. The task appears to be to predicting over distributions of cells, in which case it is worth considering a wider range of metrics that quantify prediction performance over distributions (see questions below).
- It feels that the novelty and methodological contribution is limited. SNODEP incorporates a variety of data-driven learning tools, including Neural ODEs, variational inference, and recurrent architectures. Given this, the primary contributions and novelty of the proposed method appear to be the application to predicting metabolic flux and metabolic balance in a data-driven way. In contrast, there is limited advancement in the individual methodological components used in this work.

**Questions:**

- Line 181: "the goal is to learn a model $F: t \rightarrow Y(\theta) ...$". Does the model only take time as input? My understanding is that $F$ is the decoder. Is using just time sufficient?
- What does the $Y$ variables represent? The distributions $\{G, M^f, M^b, \tilde{M}^f, \tilde{M}^b \}$? This is not entirely clear.
- If the objective is to predict entire distributions, why only consider the mean squared error (MSE) metric? Why not consider other metrics which can be specifically used to quantify the distance between distributions? For example, the Wasserstein distance and/or maximum mean discrepancy (MMD) distance.
- What is the setup for predicting metabolic flux and metabolic balance with gene knockouts, i.e. is this for *test* gene knockouts that are seen during training, or gene-knockouts *unseen* by the model during training? This is not entirely clear.

---

> ### Author Response · Authors · 2024-11-26
> **RESPONSE TO REVIEWER VVOU**
>
> We thank the reviewer for the provided insightful comments and detailed feedback. Please find our clarifications below:
>
> ---
>
> ### 1. Limited advancement in the individual methodological components used.
>
> While we agree that our framework uses several methodological components that could already be found in earlier works on an individual basis in different contexts, we believe that the sum total of those components is still novel with respect to our problem statement and biological motivation of modeling the flux-balance estimates as a random process. Please refer to our **General Response 3**.
>
> ---
>
> ### 2. Does the model only take time as input? My understanding is that $F$ is the decoder. Is using just time sufficient?
>
> We agree that we could have been clearer here. The reviewer is correct that $F$ is the trained decoder, taking $t$ and sampled latent $l_{0}, d$ as input. During inference, the input should be some prior context trajectory $(t_{i}, y_{i})_{i \in I_C}$
>  to determine the latent and $t$ for which we want to get the samples. We will clarify this in the revised version of our paper.
>
> ---
>
> ### 3. What do the $Y$ variables mean? $G, M_{f}, M_{b}, \tilde{M_{f}}, \tilde{M_{b}}$ are not clear.
>
> The $Y$ variables refer to the different types of data we consider in our modeling. Note that $Y$ is a probability distribution in our context. The different types of data are:
> - $G$: Gene-expression.
> - $M_{f}$: Metabolic-flux.
> - $M_{b}$: Metabolic-balance.
> - $\tilde{M_{f}}$: Metabolic-flux for the gene-knockout case.
> - $\tilde{M_{b}}$: Metabolic-balance for the gene-knockout case.
>
> We train a separate model for each type of data above. For each kind of data, we have samples from a particular time, which are understood as being generated by the underlying probability distributions.
>
> ---
>
> ### 4. The metrics are limited, why didn’t you try Wasserstein distance and/or maximum mean discrepancy (MMD) distance?
>
> We agree that several other metrics could have been reported as well. However, as mentioned in the subsection of *Experimental Settings* under the heading *Evaluation Metric*, we can interpret the MSE loss as the squared distance between the means of distributions, which satisfies us with the legitimacy of our metric. Additionally, another reason was the use of test-MSE as a metric in prior neural-processes papers, making it an apples-to-apples comparison with our task.
>
> ---
>
> ### 5. What is the setup for predicting metabolic flux and metabolic balance with gene knockouts?
> Is this for test gene knockouts that are seen during training, or gene knockouts unseen by the model during training? This is not entirely clear.
>
> During test time, we consider gene-knockout configurations that aren't present during training. We wanted to see whether the framework generalizes to unseen knockout configurations and unseen timesteps, and we observe that it does. This is very useful from a practitioner's point of view because a sufficiently well-trained model can generate samples for newer gene-knockout configurations, saving the time needed to perform the gene-knockout experiments.
>
> ---
>
> ### 6. Equations throughout the text don't have labels -- it would be helpful to numerically label all equations and reference them in the text.
>
> We thought the equations could be read within the context of the text around them. However, we agree that labeling them would make it easier for the reader. We will revise them accordingly.
>
> ---
>
> ### 7. The learning objective is not very clear.
>
> Our learning objective is an ELBO loss, with a slight modification where we use the context points to calculate the prior of the latent variables $L_{0}(l_{0}|\mathcal{C})$ and $D(d|\mathcal{C})$. This is similar to the loss used in the Neural-ODE Process paper.

---

> > ### Comment · Reviewer_Vvou · 2024-11-28
> >
> > Thank you for your detailed response to my many questions and for clarifying certain points! In general, I agree with the authors that this problem is important to the biological community, while also presenting interesting avenues of investigation from the side of ML research. However, regarding the current state of this work, I unfortunately remain in my position that the contribution of this work is not yet substantial enough. I believe that by incorporating some of the comments from this review and improving and clarifying the motivation (possibly through incorporating points the authors wrote in the general response), this can be a strong paper.

---

### Author Response · Authors · 2024-11-26
**GENERAL RESPONSE TO ALL REVIEWERS**

We thank all the reviewers for their insightful comments and detailed feedback. Below, we respond to the reviewers' common concerns about our work.
The remaining reviewer's questions will be clarified in each of our individual responses.

---

### 1. Why and how is the studied problem important?

Metabolic pathway analysis plays an essential role in identifying potentially novel therapeutic targets in organisms and is thus extremely important from the point of view of developing new drugs and treatments. However, gathering data about the reactions in these pathways is not easy, and as a result, people have developed genome-scale metabolic models that are used via flux balance analysis (FBA) to generate insights. For example, gene knockouts are simulated by modifying the FBA constraints and solving the optimization problem again. However, devising an accurate FBA optimization problem itself takes a lot of domain expertise, limiting its scalability. This is why we approach it from a data-driven perspective, which is a novel way to tackle the problem.

Moreover, we wanted to introduce the ML community to this very important problem. Despite the problem being motivated from a biomedical perspective, we believe it is also of significant interest from an ML perspective. For reasons (other than the ones already studied in our paper) such as:
- The presence of a graphical structure between gene-enzyme-reactions that's evolving with time.
- The potential to study gene-knockout via graph perturbations.
- Integrating both genomics and metabolomics data, leading to the generation of metabolomic data from genomic data since it's easier to procure.
- The destructive nature of single-cell sequencing reads, leading to its random process formulation with potential for better dynamical neural network models for random processes.

---

### 2. Why choose NP and NODEP as baselines?

The time-varying observations in the form of metabolic flux and balance are directly modeled as a random process, which is why we treat NP and Neural ODE processes as our baselines.

The reason we model them as a random process is that we estimate the metabolic flux and balance via scRNA-seq data, which is comparatively accessible but suffers from a destructive measurement process. This means that we cannot track the property of any individual cell (e.g., the gene-expression, metabolite concentration, or reaction flux) over time. Instead, we have samples of these quantities coming from different cells across different sampling time steps. Therefore, when we consider a metabolic-flux trajectory, the elements belong to different cells, suggesting the trajectory is a noisy one. This, in turn, implies that future values cannot be precisely predicted, and we thus predict a distribution over them.

While deep learning models aimed at extrapolating or interpolating trajectories and processing sequential data, like LSTM, GRU, or ODE-RNN, can certainly be used, getting a distribution from the predictions of a trajectory would require us to run the experiment several times, each resulting in a slightly different prediction (on account of randomness from the trajectory and the latent sampling). From these different predictions, we can estimate a mean and variance for our parametric family of choice. Neural Process family models like NP and NODEP, however, do this inherently.

---

### 3. Novelty and methodological contributions.

While we agree that our framework uses several methodological components that could already be found in earlier works on an individual basis in different contexts, we believe that the sum total of those components is still novel with respect to our problem statement and biological motivation of modeling the flux-balance estimates as a random process.

To the best of our knowledge, we haven't encountered any work that takes the time order/structure into account when modeling random processes via deep learning. While there are works like ODE-RNN or GRU-ODE that do exploit the structured-ness in the encoder, they're done in the context of interpolation or extrapolation of time-series trajectories, not random processes. Thus, we wanted to highlight that structured consideration is lacking in existing neural process methodologies, while also providing a solution for it.

---

### Meta-Review · Area_Chair_emCW · 2024-12-19

**Metareview:**

The reviewers and the authors engaged in a good discussion. Although the reviewers appreciate the problem tackled in the paper (neural ODEs for metabolic networks), the reviewers were not convinced that the paper is ready for ICLR.

The authors are encouraged to continue their work and submit the updated version of the paper elsewhere.

**Additional Comments On Reviewer Discussion:**

None.

---

### Decision · Program_Chairs · 2025-01-22

Reject